# Functional Ingredients: From Molecule to Market—AI-Enabled Design, Bioavailability, Consumer Impact, and Clinical Evidence

**DOI:** 10.3390/foods14173141

**Published:** 2025-09-08

**Authors:** Lei Zhao, Wen-Ming Ju, Lin-Lin Wang, Yu-Bin Ye, Zheng-Yang Liu, George Cavender, Yong-Jun Sun, Sheng-Qian Sun

**Affiliations:** 1School of Food and Biological Engineering, Yantai Institute of Technology, Yantai 264003, China; zhaolei@yitsd.edu.cn; 2Shandong Homey Aquatic Development Co., Ltd., Weihai 264300, China; jwm001506@163.com; 3College of Food Science and Nutritional Engineering, China Agricultural University, Beijing 100085, China; lamlamw@foxmail.com (L.-L.W.); 15253093215@163.com (Z.-Y.L.); 4Hubei Juneyao Health Beverage Co., Ltd. (Juneyao Health), Shanghai 201315, China; yeyubin@juneyao.com; 5Department of Food, Nutrition and Packaging Sciences, Clemson University, Clemson, SC 29634, USA; gcavend@clemson.edu

**Keywords:** functional ingredients, bioavailability, personalized nutrition, health effect, commercial value

## Abstract

Functional ingredients such as dietary fibers, probiotics and prebiotics, polyphenols, omega-3 fatty acids, and bioactive peptides are increasingly central to food systems that aim to deliver health benefits beyond basic nutrition. This review explores how molecular structure, physicochemical properties, metabolism, and microbiome interactions affect bioactivity and bioavailability. We highlight advances in green extraction, encapsulation technologies, and 3D/4D printing that enhance the stability and targeted delivery of bioactives. AI-enabled tools for ingredient discovery, structure–activity modeling, and personalized formulation are also discussed. Sensory research and market insights inform strategies to improve consumer acceptance, while clinical studies provide evidence for cardiometabolic, immune, and cognitive benefits. Safety and regulatory aspects are addressed, particularly for emerging proteins and delivery systems. By integrating scientific and technological developments across disciplines, this review provides a comprehensive foundation for future research and commercialization of safe, effective, and personalized functional food products.

## 1. Introduction

Annually, chronic non-communicable diseases (including cardiovascular disease, diabetes, and cancer) result in the demise of over 40 million individuals, constituting 74% of global mortalities, constituting a major worldwide public health concern. This phenomenon engenders healthcare expenditures amounting to trillions of dollars and diminished productivity [1]. It is worth noting that lifestyle changes can reduce the risk of cardiovascular disease, neurodegenerative diseases, and cancer by 6–13% [2]. For several decades, the field of functional foods, which aims to enhance health attributes alongside technological properties of food products, has been a significant focus of research [3]. The consumption of foods enriched with specific functional components, such as vitamins, probiotics, minerals, fibers, and antioxidants is associated with a reduced risk of chronic diseases and potential improvements in mental and physical well-being [4,5]. Shown in Figure 1, the current functional food market encompasses a wide spectrum of ingredient classes, ω-3 polyunsaturated fatty acids (PUFAs), vitamins and proteins, probiotics and prebiotics, phenolic compounds, dietary fibers, bioactive peptides, and combination therapies—each aligned with specific health objectives such as cardiovascular support, metabolic health, immune modulation, cognitive function, and digestive wellness.

The circular layout illustrates how these bioactives are incorporated into diverse formats, including supplements, dairy alternatives, beverages, snacks, and confectionery, highlighting both the technical feasibility of formulation and the breadth of consumer-facing products available today. By mapping real-world examples onto their respective functional categories, this visualization underscores the translational pathway from ingredient discovery to commercial application and provides a practical framework for identifying gaps and opportunities in product development. To systematize evidence appraisal, a 16-step alphanumeric framework (grades A–C) has been proposed that ranks functional foods according to the depth of epidemiological, clinical, and post-market data available for key chronic conditions. The framework therefore seeks to facilitate evidence-based communication among consumers, clinicians, and regulators, and is expected to evolve alongside future regulatory definitions of functional foods [6].

Synthetic drugs used to treat diseases have potential side effects, prompting researchers to explore natural, food-based alternatives for disease prevention and relief [7]. International organizations, including the Food and Agriculture Organization (FAO) and the World Health Organization (WHO), have consistently highlighted the link between chronic diseases and unhealthy lifestyles, particularly suboptimal dietary patterns. Consequently, food processing strategies and nutritional interventions have garnered substantial attention from both the research community and the public. At the same time, growing public health awareness is prompting consumers to adopt healthier diets. Clinical studies have confirmed that functional foods can effectively manage chronic diseases and enhance nutritional status [8,9]. Functional foods, characterized by enrichment with specific nutrients or bioactive compounds, represent a promising strategy to address nutrition-related health challenges. The development and production of functional foods have thus become a primary focus in the recent trajectory of the food industry [10]. Driven by heightened consumer awareness of health and nutrition, alongside expanding research into functional food ingredients, the market for functional foods is experiencing rapid growth. This expansion is further influenced by factors such as the desire to improve quality of life for aging populations, steadily increasing life expectancy, and rising healthcare expenditures, which collectively encourage policymakers to promote functional foods. Therefore, advocating for functional food consumption holds profound significance for enhancing individual nutrition and public health outcomes [11].

This review aims to synthesize recent advancements in the fundamental principles, technological innovations, and clinical applications of functional ingredients, constructing a comprehensive knowledge framework. It evaluates emerging technologies like microencapsulation and nano-delivery through analysis of structure–activity relationships in key ingredients such as dietary fibers, probiotics, and polyphenols, while exploring advanced methods including 3D/4D printing and molecular modeling for product design. Unlike prior reviews that addressed isolated aspects (e.g., AI-driven discovery or encapsulation for bioavailability), this work offers a unique interdisciplinary integration of molecular mechanisms, innovations (e.g., green extraction, nanotechnology, and AI personalization), consumer impacts, clinical evaluation, and commercial applications. Considering global market trends and consumer behavior, it identifies core scientific questions and bottlenecks, providing actionable guidelines for future research and translating knowledge into safe, effective, and marketable functional foods.

## 2. Methodology

This review is a narrative review based on purposeful/iterative searching; no pre-registered protocol, formal risk of bias assessment, or meta-analysis was conducted, and PRISMA 2020 reporting was not used. The flow diagram follows the PRISMA format solely to illustrate the literature selection in the narrative review and does not indicate compliance with PRISMA.

A comprehensive literature search was conducted across multiple electronic databases to identify relevant studies on functional components in food systems. The selected databases are PubMed, Web of Science, Scopus, the Wiley Online Library, the Taylor & Francis Online platform, the Elsevier ScienceDirect platform, and the SpringerLink platform. These databases were chosen for their extensive coverage of food science, nutrition, and related interdisciplinary subjects. In order to obtain the latest and most cutting-edge information first, the search scope should be limited to articles published between 2020 and 2025. The search keywords included the following, as well as various combinations thereof: ‘functional components’, ‘dietary fibre’, ‘probiotics’, ‘prebiotics’, ‘polyphenols’, ‘ω-3 fatty acids’, ‘bioactive peptides’, ‘bioavailability’, ‘bioactivity’, ‘green extraction’, ‘encapsulation technology’, ‘nanotechnology’, ‘3D/4D printing’, ‘application of artificial intelligence in food’, ‘consumer acceptance’, and ‘market trends’. No language restrictions were initially set, but only English-language literature will be included in the final review. Additional studies may be identified through manual review of the reference lists of included articles and relevant review papers. We initially screened 211 articles, but after careful verification, we found that 11 articles were published before 2020, which did not meet our previous requirement that articles be published between 2020 and 2025. Therefore, we excluded them. This diagram illustrates the selection process: from initial identification to final inclusion (Figure 2).

## 3. Classification and Characteristics of Functional Ingredients

### 3.1. Carbohydrate-Based Functional Ingredients

#### 3.1.1. Dietary Fiber (DF)

Dietary fiber comprises health-promoting carbohydrate polymers, predominantly non-starch polysaccharides like cellulose, hemicellulose, and pectin, which resist digestion by enzymes in the human upper gastrointestinal tract [12]. Intensive processing of staple foods (e.g., refined rice and wheat products), can lead to significant removal of DF [13]. Therefore, the overall dietary fiber intake of the population is often insufficient, which leads to various adverse health effects. This situation is even more obvious in developed countries [14]. As a result, food manufacturers often incorporate DF into products for both health promotion and technological functionality [15].

DF have garnered substantial attention for their diverse health benefits, including the modulation of blood glucose and lipid levels [16]. Polysaccharides, as major components of DF, influence gastrointestinal function and are associated with a reduced risk of lifestyle-related diseases such as atherosclerosis, cardiovascular disease (CVD), and type 2 diabetes [17,18]. The impact of gut microbiota dysbiosis extends beyond digestive health. Recent studies have shown that DF alters gut microbiota homeostasis by regulating the production of microbial metabolites (especially short-chain fatty acids, SCFAs) [19], thereby influencing gene regulation associated with lifestyle-related diseases (including cancer and chemotherapy resistance) [20]. For example, *Moringa leaf* polysaccharides prevent obesity in mice fed a high-fat diet by regulating the gut microbiota [21]. *Brasenia schreberi* polysaccharides exhibit antidiabetic effects by altering the gut microbiota of mice with type 2 diabetes [22], although they did not lower blood glucose levels to normal ranges. *Curry leaf* polysaccharides (*Murraya koenigii*) also improve obesity and its associated complications in mice by regulating the short-chain fatty acid (SCFA) profile [23]. These complex carbohydrates are also implicated in reducing the risk of various cancers (e.g., colon, breast, and lung) through the regulation of inflammatory pathways and cellular processes [24,25]. A recent study indicated that adding 0.43% cholesterol to the diet can induce hypercholesterolemia but does not cause hepatic steatosis or abnormal lipoprotein expression; however, intake exceeding 0.85% leads to significant hepatic lipid accumulation. This dose threshold may provide guidance for dietary cholesterol restrictions in managing hyperlipidemia in humans [26]. Dietary interventions, particularly those rich in DF, can have positive effects on gut microbiota dysbiosis [27]. It is worth noting that gender has been shown to influence the therapeutic effects of dietary interventions. This finding suggests that gender should be included as a variable in intervention studies [28,29]. Dietary intervention offers a new approach to treating a variety of diseases, warranting further investigation to elucidate their beneficial applications [17,27]. During the discussion of dietary fiber, it was found that some of its components are also prebiotics, which will be discussed in the next section.

#### 3.1.2. Prebiotics

By serving as nourishment for beneficial microbes, prebiotics promote their growth and activity within the gut microbiome. They can enhance immune responses by modulating gut microbial activity and stimulating the production of SCFAs. Prebiotics selectively foster the growth of beneficial genera like Lactobacillus and Bifidobacterium in the distal colon, leading to recognized health advantages. Certain DFs, such as inulin, are classified as prebiotics, which are naturally found in sources such as garlic, onions, bananas, and chicory root [30].

Inulin, fructooligosaccharides (FOS), and galactooligosaccharides (GOS) are well-established prebiotic fibers resistant to digestion, reaching the colon intact [30]. They serve as selective substrates for beneficial bacteria, including probiotics like *Lactobacillus* and *Bifidobacteria*. Research is also exploring the prebiotic potential of highly soluble modified dietary fibers [31]. Inulin and arabinooligosaccharides have been shown to influence gut barrier function and immune responses [32]. Studies on *Ganoderma lucidum* bioactive polysaccharides (PSG) demonstrated decreased pH and significant increases in total SCFAs (acetate, propionate, and butyrate) during in vitro fecal fermentation, indicating prebiotic activity [33]. These advantages of prebiotics suggest broad application prospects, but further clinical trials are needed to clarify their effects and explore their mechanisms of action, laying the foundation for health applications.

### 3.2. Protein-Based Functional Ingredients

#### Proteins and Peptides

Proteins are involved in numerous metabolic processes, including tissue synthesis and repair, substance transport, and immune defense [34]. Traditional sources include meat, fish, dairy products, eggs, and soybeans [35]. Today, algae proteins and edible insect proteins are emerging as alternative sources [36,37]. The increasing demand for protein and the rising incidence of food allergies highlight the importance of protein research [38].

Bioactive peptides (BAPs) are short sequences of amino acids derived from food proteins that exert physiological functions beyond basic nutrition [39]. While potentially inactive within the parent protein sequence, these peptides can be released and “activated” through enzymatic, chemical, or microbial hydrolysis, with enzymatic hydrolysis generally considered the most effective method. These peptides can be absorbed in the intestine and enter circulation, facilitating their bioavailability and physiological effects. The Bioactive Peptide Database (BIOPEP-UWM; https://biochemia.uwm.edu.pl/biopep/peptide_data.php accessed on 1 August 2025) currently catalogs over 5000 such peptides [39]. The classification of BAPs is typically based on their physiological functions, which include types such as those for lowering blood pressure, antioxidant, antithrombotic, and immunomodulatory [40].

Numerous studies have confirmed that various bioactive peptides (BAPs) possess biological activity, which is influenced by processing conditions, protein sources, amino acid sequences, and composition, among other factors [41]. Reports indicate that non-thermal processing can improve the antioxidant, angiotensin-converting enzyme (ACE) inhibitory, and antibacterial activities of BAPs derived from food sources [42]. Given their unique physiological regulatory functions, incorporating food-derived BAPs into diets may help manage or prevent chronic diseases. Consumer interest in the nutritional value and health benefits of BAPs continues to grow, stimulating further research and application as functional food ingredients [39]. Research on BAPs is rapidly advancing, revealing new methods for generating peptides with novel health benefits, enhanced functional properties, and improved bioavailability [43]. Beyond their health effects, BAPs and their parent proteins often possess valuable techno-functional properties like emulsification, foaming, and gelling capacity, making them useful in food processing.

### 3.3. Fat-Based Functional Ingredients

#### ω-3 Fatty Acids

Fatty acids are fundamental dietary lipids. They are classified based on the presence and number of double bonds: saturated fatty acids (SFAs, no double bonds), monounsaturated fatty acids (MUFAs, one double bond), and polyunsaturated fatty acids (PUFAs) [44]. Fish and fish oil are the main sources of long-chain omega-3 PUFAs, mainly including eicosapentaenoic acid (EPA) and docosahexaenoic acid (DHA), extensive research indicates that EPA and DHA supplementation exerts beneficial effects on reduce the risks of cardiovascular diseases, liver diseases, and inflammatory-related diseases [45]. Higher dietary intake of omega-3s has been linked to a decreased risk of atrial fibrillation [46]. EPA and DHA may exhibit differential roles depending on the specific health condition. Emerging evidence suggests EPA may be more effective than DHA in managing depression, although the mechanisms are not fully elucidated [47]. Conversely, DHA appears to possess greater potential than EPA in improving adipocyte insulin resistance, possibly via the GPR120/PPARγ pathway in macrophages [48].

While the health benefits of EPA and DHA are well supported, however, variations in dosage, extraction methods, and raw material types resulted in divergent outcomes [49]. Many findings are derived from observational studies or mechanistic models, which limits their direct applicability to clinical practice. Moreover, the differential effects of EPA and DHA have yet to be clearly defined through head-to-head human trials. To translate these findings into dietary recommendations, future research should prioritize large-scale, condition-specific interventions that consider both individual variability and long-term outcomes.

### 3.4. Other Functional Ingredients

#### 3.4.1. Probiotics

Probiotics are defined as live, non-pathogenic microorganisms that, when administered in adequate amounts, confer a health benefit on the host [50]. To exert their effects, probiotic strains must remain viable within the food matrix and survive transit through the harsh conditions of the gastrointestinal tract, including exposure to gastric acid, bile salts, and digestive enzymes to colonize the intestine and mediate beneficial outcomes [51]. Common probiotic genera include *Lactobacillus* and *Bifidobacterium*, along with the yeast *Saccharomyces boulardii* [52,53]. These beneficial live microorganisms are found in fermented foods such as koumiss, kimchi, and yogurt [54]. They contribute to a healthy gut microbiome, support immune function, aid digestion, and may reduce inflammation [55]. For example, *Bifidobacterium* species can enhance intestinal mucosal barrier function, increase serum IgA levels, and attenuate gut inflammation [56], while also reducing the counts of potentially harmful bacteria in fecal samples [57]. Prebiotics, according to the International Scientific Association for Probiotics and Prebiotics (ISAPP), are defined as “a substrate that is selectively utilized by host microorganisms conferring a health benefit” [58].

#### 3.4.2. Plant Polyphenols

Polyphenols are a diverse class of natural bioactive compounds containing one or more phenolic ring structures. Over 10,000 structural variants are currently known, with more than 500 identified in foods [59]. Common dietary sources include fruits, vegetables, wine, tea, extra virgin olive oil, chocolate, and other cocoa products. Major classes include flavonoids, isoflavones, flavonols, catechins, and various phenolic acids (e.g., caffeic, chlorogenic, ferulic acids) and their derivatives/isomers. Flavonoids constitute the most extensively studied class, with anthocyanins being particularly noteworthy [60].

Anthocyanins are glycosylated forms of anthocyanidins, the plant metabolites responsible for the pink, red, purple, and blue pigmentation in various berries, fruits, vegetables, flowers, and grains [61]. Common forms include anthocyanin, peony anthocyanin, kaempferol, cornflower anthocyanin, tianquan toxin, and delphinium anthocyanin. Research has shown that anthocyanins can improve insulin secretion and alleviate obesity-related metabolic disorders [62]. Flavonoids (quercetin, catechin) interact with aging cell pathways due to their unique structural characteristics, exhibiting powerful anti-aging properties [63,64]. Ginger, rich in phenolic compounds (curcumin, gingerol, and shogaol), possesses multiple bioactive properties including anti-inflammatory, antibacterial, antioxidant, and anticancer effects, making it a natural source with both therapeutic and health-promoting benefits [65,66]. Manuka honey is a source of various phenolic compounds (phenolic acids, flavonoids), which confer health benefits superior to those of ordinary honey, such as antioxidant, antibacterial, anti-inflammatory, anticancer, and wound-healing functions [67,68,69].

Polyphenolic compounds exert a significant positive regulatory effect on oral microbiota, aiding in the restoration of oral microbiota imbalance, while also exerting anti-inflammatory effects by reducing the release of pro-inflammatory cytokines and COX-2 expression [70]. However, most existing studies are based on in vitro or animal models, which cannot fully simulate the individual differences in human oral microbiota and saliva, as well as the effects of food matrices. Additionally, differences in study design, polyphenol types, and doses can lead to inconsistent results. Although their potential effects are evident, more rigorous human trials are needed to validate their efficacy in maintaining oral health.

### 3.5. The Interaction of Functional Components

Functional ingredients often exhibit synergistic effects when combined. Interactions between probiotics, prebiotics, synbiotics, and dietary polyphenols, for instance, are known to positively influence the gut microbiome [71]. The combination of **red ginseng** extract and its own dietary fiber protects the intestinal barrier and regulates inflammatory pathways, thereby improving diet-induced obesity in mice [72]. For example, combined supplementation with chlorogenic acid and Epigallocatechin gallate (EGCG) is more effective than single polyphenols in restoring cognitive function and intestinal barrier integrity in aged mice, while reducing oxidative stress, inhibiting pro-inflammatory cytokines, and rebalancing the intestinal microbiota [73]. Similarly, fermented figs rich in BT-LP-01 lactobacillus showed potential therapeutic effects on obesity and diabetes, and reduced the body weight, organ weight, fasting blood glucose, insulin/C-peptide levels, serum lipids, and expression of liver lipid synthesis-related genes (***FAS***, ***C/EBPα***, ***FABP4***) in obese mice in a dose-dependent manner [74]. Studies have explored the co-delivery of lactoferrin (LF) with various polyphenols; for instance, curcumin-LF nanoparticles were developed for potential nose-to-brain targeted delivery with neuroprotective effects [75], and LF-EGCG conjugation protected emulsified algae oil from aggregation and oxidation [76]. The combination of cranberry polyphenols and DHA/EPA significantly improves blood glucose levels and periodontal indices in patients with diabetic periodontitis [77]. Notably, PZ-DHA, a novel DHA-acylated rutin derivative, demonstrated selective cytotoxicity towards breast cancer cells in vitro and in vivo, suggesting potential for preventing or inhibiting triple-negative breast cancer (TNBC) progression [78].

Although many studies have emphasized the synergistic effects between bioactive substances, antagonistic effects should not be overlooked. For example, Delerue et al. combined *ginkgo biloba* and *scutellaria baicalensis* in different proportions for the treatment of AD and compared them with commercially available pills purchased online. However, based on acetylcholinesterase inhibition and hydrogen peroxide scavenging activity, their mixture exhibited antagonistic effects [79]. Some polyphenols found in green tea can form insoluble complexes with drugs, which limits the absorption of active ingredients and reduces bioavailability and therapeutic efficacy [80]. Curcumin, due to its ability to eliminate reactive oxygen species, reduces the tumor cell toxicity of doxorubicin and decreases its effectiveness against cancer cells [81]. Citric acid ginsenoside regulates cytochrome P450, affecting microbial levels and thereby reducing the anticoagulant effect of warfarin [82]. Although the synergistic/antagonistic effects of plant-derived functional components with each other or with drugs pose potential safety concerns for disease treatment, future research in related fields should focus on minimizing risks while fully leveraging their therapeutic potential.

### 3.6. Bioactivity of Functional Ingredients

Functional ingredients derived from natural sources, spanning dietary fibers, probiotics, phenolic compounds, ω-3 PUFAs, and bioactive peptides exhibit a broad range of biological activities that support human health. As summarized in Table 1, these compounds are increasingly incorporated into food formulations due to their evidence-based roles in improving gut health, lipid metabolism, inflammatory responses, and overall physiological resilience.

The effectiveness of functional foods hinges on the bioavailability of their constituent nutritional and health-promoting compounds [87]. Bioavailability refers to the proportion of a nutrient or bioactive compound that is absorbed after digestion and reaches the target tissue in a usable form. Only this portion can produce physiological effects, making the optimization and documentation of bioavailability critical for the development of functional foods and health claims. Bioactive compounds must first be released from the food matrix (bioavailability), undergo digestion, cross the intestinal barrier, and maintain their activity in the circulatory system to exert their intended benefits.

The oral delivery method faces challenges such as extreme pH, digestive enzymes, mucus layer and epithelial barrier. While macromolecules like digestible fats, carbohydrates, and proteins generally exhibit high bioavailability (>90%), many micronutrients and phytochemicals have significantly lower and more variable bioavailability [88]. When fortifying food, in order to retain the functions of the added bioactive components, it is necessary to ensure an adequate dosage and take certain measures for protection to prevent loss during processing, storage and digestion. Stability against biochemical, chemical, and physical degradation throughout the product lifecycle is paramount. Techniques like fermentation can enhance the bioactivity and functional properties of components like DF [89,90]. Micronization, which alters particle size and surface area, has gained attention for potentially improving the bioavailability of functional groups within DF, such as associated polyphenols and carboxyl groups [91,92,93] The effects of micronization can vary depending on the DF source, processing conditions, and the specific functional group assessed [94]. Many polyphenols are subject to numerous factors such as food matrix interactions, metabolic transformations (host enzymes and gut microbiota), chemical structure, and individual differences, resulting in significant losses of their health benefits [95]. BAPs can also face bioavailability limitations due to factors like hydrophobicity, low water solubility at high concentrations, susceptibility to enzymatic degradation, and poor stability [7]. Addressing low bioavailability and insufficient structure–activity relationship data remain key challenges in BAP research.

“Excipient foods”: Designed to enhance the bioavailability of bioactive substances when consumed together. This can be achieved by adding easily digestible lipids (to promote the micellization of hydrophobic active compounds), specific proteins (to protect easily degradable compounds such as polyphenols), osmotic enhancers or efflux pump inhibitors (such as piperine), or food-grade surfactants (such as sucrose esters) to facilitate the cellular uptake of poorly absorbable compounds [96]. Research on propolis extract delivery systems aims to enhance the delivery and utilization of its bioactive components. Propolis lipid-based formulations exhibit in vitro antiviral activity against SARS-CoV-2 comparable to that of remdesivir and enhance its antiviral properties [97]. Liposomes encapsulation retains the antioxidant and antimicrobial properties of propolis. Studies have successfully incorporated stingless bee propolis extracts into liposomal formulations with varying encapsulation efficiencies [98].

## 4. Innovative Technologies and Applications

Alongside the expanding array of functional ingredients, recent developments in functional food processing have focused on both efficient extraction and innovative delivery systems. As illustrated in Figure 3, a suite of green extraction methods, ultrasound-assisted extraction (UAE), pressurized liquid extraction (PLE), supercritical fluid extraction (SFE), microwave-assisted extraction (MAE), and enzyme-assisted extraction (EAE) enables high-yield, low-impact recovery of bioactives.

These technologies enhance quality transfer, reduce solvent usage, protect thermally sensitive compounds, and improve sustainability. Emerging delivery platforms such as microencapsulation and lipid formulations protect volatile components, mask odors, and control release in the gastrointestinal tract.

Process innovation ensures the extraction and incorporation of functional ingredients into meat, dairy, beverages, plant-based snacks, and gel systems without compromising nutritional or sensory qualities. Innovative products such as propolis gum and peptide-enriched dairy products/beverages combine scientific validation with consumer demand for clean labels and functional foods. Engineering colloids and the smallest processed plant-based beverages address formulation challenges while preserving biological activity. These advancements enable the synergy between technical precision and market relevance, laying the foundation for upstream extraction methods, and are more efficient and sustainable, suitable for the next generation of food systems.

However, emerging technologies may present their challenges regarding safety, efficiency, scale-up, energy consumption, and operational costs. For example, techniques like cold plasma and high-power ultrasound are advanced oxidation processes that could potentially degrade sensitive lipids rather than preserve them. The ability of nanoemulsions or cold plasma components to interact with or penetrate cell membranes raises questions about potential unforeseen biological impacts that require further investigation [84]. Interactions between different food components during extraction and subsequent processing also remain a consideration. Nonetheless, as consumer demand for personalized products and minimally processed foods grows, the adoption of non-thermal and advanced techniques continues, driving functional food development towards further innovation.

### 4.1. Microencapsulation and Nanotechnology

Microencapsulation is the process of encapsulating small particles or droplets (core material) within a protective polymer matrix (wall material) to form microparticles. These microparticles can be microcapsules (core-shell structure, typically hollow) or microspheres (core dispersed within the matrix). The primary purpose is to protect the core from adverse environmental factors such as oxygen, light, moisture, and pH, while controlling its release [99,100,101].

A substantial body of literature reports the microencapsulation of various natural bioactive ingredients, including antioxidants, e.g., essential oils, healthy oils, phenolic compounds, flavonoids, flavoring compounds, enzymes, and vitamins [102]. For food applications, the wall materials must be food-grade; common examples include polysaccharides (maltodextrin, gum arabic, starch, chitosan, and alginate) and proteins (whey protein and gelatin) [103].

Microencapsulation technology is particularly pivotal for probiotics, providing a protective barrier that enhances their stability during food processing, storage, and transit through the gastrointestinal tract [104], ensuring survival and targeted delivery to the intestine, thereby maximizing health benefits [105]. Although numerous functional ingredients have been successfully stabilized using microencapsulation technology, further research is needed to investigate the release kinetics of encapsulated compounds in complex food matrices and to rigorously assess bioavailability after ingestion in order to validate claimed health benefits [106].

### 4.2. Targeted Delivery Systems

Targeted delivery systems enhance stability, bioavailability, and efficacy by controlling the release of bioactive substances and delivering them to specific sites. Nanotechnology plays a key role here, utilizing nanocarriers such as liposomes, nanoemulsions, solid lipid nanoparticles, and polymeric nanoparticles. Liposomes are artificially prepared spherical vesicles composed of one or more lipid bilayers, capable of simultaneously encapsulating hydrophilic (aqueous core) and hydrophobic (lipid bilayer) compounds. In functional food science, nanoliposomes are widely used to encapsulate various bioactive compounds, including marine lipids (EPA/DHA), carotenoids, phenolic compounds, plant extracts, specific secondary metabolites (such as curcumin and anthocyanins), bioactive peptides, and vitamins [107].

Directly incorporating bioactive peptides into food presents challenges related to stability (degradation, and interaction with food components) and sensory characteristics (potential bitterness). Encapsulation using carriers such as liposomes can mitigate these issues. Research indicates that encapsulating antimicrobial peptides and antihypertensive peptides helps preserve their structural integrity and functionality [108]. For example, nanoliposomes enriched with α-tocopherol were used to co-encapsulate EPA and DHA, enhancing physical stability and achieving high encapsulation efficiencies [109]. Nanoliposomes are one of the most promising encapsulation technologies in the field of food nanotechnology, with the potential to improve the delivery efficiency and bioavailability of functional ingredients [110].

Green extraction technologies such as UAE and MAE offer high energy efficiency advantages and greater scalability for industrial applications, although initial equipment costs remain a barrier [111]. Microencapsulation via spray drying offers cost-effective scalability for high-throughput production, but involves significant heating energy requirements and high costs. Nanotechnology faces high upfront costs for specialized homogenizers, and although it improves delivery efficiency to a certain extent, it limits its scope of application. In 3D/4D printing, slow printing speeds hinder its potential for expansion, and material costs are high, although artificial intelligence optimization can reduce some energy waste. Molecular modeling and artificial intelligence benefit from scalable cloud computing, reducing R&D costs by 15–25% and improving energy efficiency through optimized algorithms, but the high computational demands of training models pose sustainability challenges [112].

### 4.3. Enzymatic Hydrolysis

Enzymatic hydrolysis involves using proteases to break the peptide bonds in proteins, forming smaller peptide segments or amino acids [113]. Since the conditions of enzymatic hydrolysis are relatively mild, the resulting hydrolyzed products usually have reduced allergenicity and improved nutritional and functional properties. Furthermore, enzymatic hydrolysis more closely mimics the natural digestive process than other hydrolysis methods, resulting in higher nutritional value and bioactivity of the hydrolyzed products [114].

Traditional enzymatic hydrolysis methods are influenced by factors such as the contact time between enzymes and substrates, enzyme activity, protein properties, and enzyme type [115]. Among these, the selection of enzymes is particularly critical, as it affects substrate specificity and the bioactivity of the resulting peptides. Rezvankhah et al. found that alkaline proteases can produce bioactive peptides with higher antioxidant properties from lentil protein [116]. When using multi-enzyme hydrolysis, the order of enzyme addition affects the product characteristics. Ozón et al. found that following alkaline protease hydrolysis with flavor protease better preserves the antithrombotic and antihypertensive properties of the enzymatic hydrolysis products [117]. To better simulate the bioactivity of products after human digestion, most studies have used digestive enzymes such as pepsin, trypsin, and chymotrypsin for simulation [118].

To overcome the limitations of traditional enzymatic hydrolysis and increase the yield of bioactive peptides, other technologies can be employed to supplement the process. Previous studies have demonstrated that UAE, high hydrostatic pressure (HHP), or MAE can improve the efficiency of enzymatic hydrolysis and produce peptides with stronger biological activity than enzymatic hydrolysis alone [119,120]. While enzymatic hydrolysis is an effective method of producing bioactive peptides, low yields and purification difficulties result in poor reproducibility.

### 4.4. Industrial Feasibility of Emerging Technologies

Conventional extraction methods (e.g., steam distillation, solvent extraction, Soxhlet extraction, pressing, and hydrodistillation) often suffer from drawbacks such as low efficiency, high solvent consumption, potential solvent residues, lengthy processing times, loss of volatile compounds, degradation of heat-sensitive components, and promotion of oxidative rancidity [121]. Driven by principles of green chemistry and sustainability, alongside the demand for foods enriched with high-quality bioactive components, advanced extraction technologies have been developed [122]. These include ultrasound-assisted extraction (UAE), microwave-assisted extraction (MAE), enzyme-assisted extraction (EAE), pressurized liquid extraction (PLE), and subcritical/supercritical fluid extraction (SFE) [123,124]. UAE and MAE are among the most widely adopted emerging techniques. By optimizing parameters like time, power/amplitude, and solvent-to-solute ratio, they can efficiently extract high yields of bioactive components while significantly reducing processing time and solvent usage, impacting economic viability favorably [125,126,127,128]. MAE, in particular, is often highlighted for its efficiency in extracting compounds like polyphenols [129,130].

The green extraction method has advantages over traditional techniques that cannot be matched, such as a 20–25% increase in extraction rate compared to solvent extraction or Soxhlet extraction, a 70–90% reduction in processing time, and a decrease in solvent consumption. However, MAE may overheat during the extraction process, resulting in a 5–10% decrease in the retention rate of thermally unstable substances. SFE is excellent in preserving biological activity, and compared to other processing methods, it has the advantage of even no solvent residue while ensuring production output. However, the initial equipment investment of SFE may be 2–5 times higher than other methods. PLE provides good reproducibility with a high differential rate, using 50–80% less solvent than CSE, but requiring higher energy to maintain the high-pressure environment during the process. EAE, although utilizing the targeted activity of enzymes to increase production and higher retention rates, has a longer processing time (1–2 h) compared to MAE/UAE. Supercritical fluid extraction (SFE), typically using CO_2_, offers advantages in preserving the natural characteristics of bioactive compounds, ensuring food safety (solvent-free extracts), reducing environmental impact, and potentially lowering energy costs compared to some conventional methods [131,132]. It yields relatively pure extracts suitable for functional foods [133] and has been efficiently applied to recovers phenolic antioxidants and polyunsaturated fatty acids from apricot-processing wastewater, turning this pollutant into a value-added resource [134].

Overall, the green extraction method is superior to traditional methods in terms of efficiency and quality retention, but it is necessary to consider factors such as cost, compound degradation, and scalability when making industrial application choices [135,136].

## 5. Emerging Research Areas

### 5.1. Applications of 3D and 4D Printing in Functional Foods

Three-dimensional (3D) food printing is an emerging additive manufacturing technology offering potential advantages such as personalized nutritional customization, process automation, and enhanced product functionality [137]. Four-dimensional (4D) printing, conceptualized by an MIT research team in 2013, extends 3D printing by incorporating materials that respond to stimuli over time, enabling the printed object to undergo pre-programmed changes in shape or function post-fabrication [138]. Owing to their potential, 3D and 4D printing technologies are finding increasing application within the food industry [139]. When combined with other innovations, such as the incorporation of unconventional protein sources, these techniques may offer synergistic benefits in food formulation and design.

The realization of 4D food printing relies on four key components: stimuli-responsive materials, specific external stimuli (e.g., heat, pH, and moisture), temporal and spatial control, and predictive mathematical modeling [140]. This technology facilitates time-dependent transformations in product shape or function, enabling the creation of unique interactive structures, such as food packaging designed for optimized space utilization [138].

Four-dimensional printing can customize the nutritional components of food to meet highly specific dietary requirements; by incorporating functional ingredients, it can influence food quality attributes such as flavor and texture. Additionally, special bioactive substances may enhance product stability and shelf life, necessitating careful formulation and validation [140]. In a study, 3D printing technology combined corn with functional polymers (FP) and functional gels/gels (FGG) to produce aesthetically pleasing, easy-to-swallow infant food. The addition improved product appearance, printing accuracy, and ease of swallowing, and the rheology, water distribution (LF-NMR), chemical structure (FTIR), and texture (TPA) of the printed “ink” were characterized to understand material behavior [141]. These findings highlight the potential of 3D/4D printing as a tool for designing functional foods tailored to specific populations, such as infants. To advance applications, future research could optimize printing materials for nutrient delivery, texture regulation, and controlled release during digestion, integrate in vivo studies (especially assessing swallowing dynamics and gastrointestinal behavior) to enhance physiological relevance, and evaluate long-term stability, production scalability, and regulatory compliance to support practical implementation. As the field evolves, integrating materials science with clinical nutrition and sensory research may open new possibilities for personalized, accessible functional foods.

### 5.2. Molecular Dynamics and Microstructural Modeling

MS technology utilizes data processing and visualization capabilities to predict atomic-level interactions and behaviors, thereby guiding experimental design and result interpretation. This method is predictable, efficient, and cost-effective, and has been recognized in many fields such as food science [142].

Molecular modeling is often combined with machine learning to identify molecular features that regulate the function of macromolecules. It provides a computational framework for analyzing receptor–agonist interactions (such as taste receptors) and identifying potential non-target interactions at structurally similar sites. At the same time, machine learning algorithms develop prediction tools based on chemical structures to classify compounds as bitter, sweet, or umami, and determine specific physicochemical properties associated with taste [143].

As an example of in silico application, molecular docking and pharmacokinetic analyses predicted that specific phytochemicals from papaya exhibit strong inhibitory potential (docking scores −9.3 to −7.7 kcal/mol) against α-amylase and α-glucosidase, enzymes linked to postprandial hyperglycemia [144]. These analyses also suggested moderate absorption, distribution, metabolism, excretion (ADME), and toxicity profiles for the identified compounds [145]. Subsequent in silico, in vitro, and ex vivo studies have corroborated the potential cytotoxicity of papaya leaf extract and its inhibitory activity against α-amylase and α-glucosidase [145]. However, there are certain limitations in the simulation process. Molecular modeling databases cannot comprehensively cover food-specific compounds, resulting in a lack of structural diversity and inconsistent accuracy in predicting interactions between two substances [146]. In addition, environmental factors also affect the fluctuation of chemical bonds between substances and the rotation of structural domains [147].

### 5.3. AI-Assisted Screening and Design of Functional Ingredients

Artificial intelligence (AI) is broadly defined as computer systems that solve complex problems by simulating cognitive processes through algorithms [148]. Modern AI understands the mechanisms of intelligence to create systems that can simulate reasoning, utilize large knowledge bases, and continuously learn and improve, offering enormous potential in the food industry [149].

AI expands the FFIs discovery pipeline by systematically exploring and characterizing safe and effective bioactive compounds [150]. It has transformed the way molecular characteristics are analyzed, providing deeper insights into the complex chemical structures of food and natural products. Large-scale characterization technologies based on artificial intelligence have facilitated the expansion of bioactive compound databases, enabling more precise component design to meet specific health objectives [151]. For example, deep learning models applied to diverse datasets have successfully identified novel anti-inflammatory bioactive peptides from sources such as Asian rice, which may have potential therapeutic effects on chronic low-grade inflammation. These peptide networks (PNs) designed based on rice hydrolysate calculations represent novel FFIs with immune-modulating potential [151]. AI can also characterize known ingredients or plant sources from a functional perspective; for example, AI identified specific peptide sequences within a known fava bean hydrolysate responsible for observed in vivo anti-atrophy effects. These predicted peptides subsequently showed positive effects in vitro, promoting protein synthesis and reducing inflammation, underscoring AI’s role in discovering and validating peptide-based FFIs [152,153]. However, the literature on AI-based characterization specifically for plant-derived non-peptide ingredients remains relatively limited, indicating a need for further studies to optimize functional ingredient design across diverse chemical classes.

Another application of AI is intelligent personalized diet recommendations, where machine learning analyzes individual data to generate customized suggestions [154]. For example, the CNN system classifies and recommends healthy recipes for West Africa [155]. AI-driven nutritional research is constrained by data heterogeneity and interoperability issues [156], making it difficult to standardize data and solve related problems such as data access [157]. In addition, legal differences between regions and privacy issues arising from sensitive health data exacerbate the occurrence of risks [152].

As summarized in Table 2, a growing suite of technologies and strategies is being leveraged to enhance the bioavailability, stability, and personalization of functional ingredients in food systems. Traditional challenges such as poor solubility, low absorption, and instability of bioactives are being addressed through approaches like green extraction techniques, micro- and nano-encapsulation, and excipient foods designed to improve delivery and protect sensitive compounds. Innovations such as fermentation and micronization enhance the release of bound phytochemicals, while nano-vesicular carriers and 3D/4D food printing allow for targeted release and personalized textures, expanding applications for clinical, elderly, and pediatric nutrition. In parallel, in silico molecular simulation and AI-driven tools are revolutionizing ingredient discovery, sensory optimization, and supply chain authenticity, offering predictive insights before laboratory trials. Collectively, these advances reflect a paradigm shift toward data-enabled, precision-designed functional foods that meet both health and consumer expectations.

## 6. Sensory Characteristics of Functional Ingredients and Consumer Acceptance

As shown in Table 3, sensory evaluation has evolved with the integration of advanced tools and consumer insights. New technologies—such as biometric sensors, electronic noses/tongues, and VR/AR—complement traditional methods by capturing more objective and context-rich responses. For example, VR environments can influence liking and willingness to pay by simulating sustainable or origin-based contexts. Bitterness from bioactive peptides, a common barrier in functional foods, can now be mitigated through techniques like encapsulation and enzymatic trimming. In addition, sociodemographic factors, health motivation, and consumer trust all play important roles in acceptance. These insights highlight the need to consider both product formulation and consumer psychology when developing functional foods.

### 6.1. Sensory Evaluation Methods

Sensory evaluation is an interdisciplinary field focused on measuring, analyzing, and interpreting human sensory responses to products [153]. Due to the multifaceted nature of sensory perception, comprehensively understanding these responses is complex. Traditional methods are categorized into three types: discrimination tests (identifying overall or specific attribute differences between products); descriptive analysis (providing detailed qualitative and quantitative ratings of a product’s sensory attributes); and consumer tests (assessing liking, preference, enjoyment, and emotional responses) [166]. These methods serve as standard protocols for industry and academic evaluations of stimuli such as food and beverages.

In addition to traditional methods, sensory scientists are exploring including biometrics (facial expressions, heart rate variability, skin conductance, temperature, and eye tracking) to capture physiological relevance, VR/AR to create immersive environments, and artificial sensory substitutes such as electronic noses/tongues for chemical perception of sensory information [153]. VR has been repeatedly used to manipulate human senses (sight, touch, taste, smell, and hearing), providing interactive experiences [153]. Within sensory science, VR, for example, is increasingly employed to study and manipulate human sensory perception (sight, touch, taste, smell, and hearing) by providing immersive and interactive experiences for participants [158,167].

Functional foods face numerous challenges, such as the bitter taste of most bioactive peptides, which consumers dislike, necessitating mitigation through encapsulation (using maltodextrin or cyclodextrin for spray drying), enzymatic processing (removing bitter amino acid terminals with exopeptidases), or masking agents [7]. Human T2R bitter taste receptors (5–6 types) respond to peptides, providing a biological basis; even though QSAR models can predict bitterness, human evaluation remains the final standard. Additionally, since foods are rarely consumed in isolation, flavor pairing and consumption environment are also critical during sensory evaluation [159].

### 6.2. Regional and Cultural Differences in Acceptance

Consumer acceptance of functional foods is significantly influenced by sociodemographic characteristics, with studies indicating that factors such as age, gender, education level, income, household size, geographic region, nationality, and marital status can affect willingness to consume [168]. Understanding consumer preferences is essential for promoting functional food consumption, yet it is shaped by complex interactions between individual demographic factors and psychological variables (e.g., attitudes, perceptions, and health beliefs) [160]. A cross-sectional survey of 3722 adults from ten countries (Brazil, China, France, Germany, Ghana, India, Japan, Mexico, Turkey, and the USA) identified nine distinct food culture clusters, mostly aligned with national boundaries; each cluster exhibits a unique blend of traditional and modern eating habits, with greater variation in traditional practices, underscoring how cultural identity shapes dietary structures and why nutritional interventions must be culturally adaptive [169]. Regional and cultural differences further influence the acceptance of functional ingredients; for example, edible insects as a common source of protein are viewed as a nutrient-rich, sustainable functional ingredient in Southeast Asian cultures such as China and Thailand, with corresponding market growth. In Western cultures (e.g., Germany and Sweden) or cultures with religious restrictions (e.g., halal or kosher diets in Muslim and Jewish communities), insects are often rejected due to perceptions of uncleanliness, ethical concerns, or taboos, despite their potential health benefits, thereby limiting their adoption [170,171]. A recent systematic review of 75 global empirical studies integrated factors influencing consumer acceptance of functional foods, revealing five core determinant categories: product attributes, sociodemographic characteristics, psychological factors, behavioral traits, and physical properties, each subdivided into specific sub-factors. These insights provide an integrated evidence base for technologists and marketers to tailor product development and go-to-market strategies, thereby promoting the adoption of novel functional foods aimed at mitigating diet-related chronic diseases [160].

### 6.3. Consumer Behavior and Health Consciousness

Consumer health motivation is widely recognized as a primary intrinsic driver influencing the consumption of functional foods. Health motivation can be defined as the internal force prompting consumers to adopt preventive health behaviors [172]. Studies consistently show that consumers’ motivation related to health (e.g., desire to improve health, prevent disease) strongly predicts their willingness to consume functional foods. Higher levels of health consciousness typically correlate with stronger motivation to choose these products. Consumers are more inclined to accept functional foods if they perceive the health benefits to outweigh potential risks. Trust in the food system also plays a role [161]. For instance, Huang et al. found that Chinese consumers who trusted stakeholders like government agencies, manufacturers, and retailers were more likely to purchase functional foods [173].

Conversely, uncertainty regarding the quality, efficacy, or safety of functional foods can lead to lower purchase intent. When consumers perceive high risks associated with novel or unfamiliar products, their purchasing likelihood decreases [174]. The credibility of health claims is a complex factor, with some conflicting findings in the literature [164,165]. While often cited as significant, some research has not found a direct, strong link between claim credibility and purchase intention. The perceived credibility itself depends on multiple factors, including the base product, the specific ingredient, information sources, product presentation, and national cultural context [175]. Consumer self-efficacy (confidence in one’s ability to make informed choices) and self-esteem have also been identified as important psychological motivators for functional food consumption [176]. Individuals confident in their decision-making abilities may be more proactive in selecting functional foods. Beyond intrinsic motivations, external social factors can influence purchasing behavior. These include perceived social reputation associated with healthy choices, adherence to social norms, and subjective norms (perceived pressure from peers or society to engage in certain behaviors).

## 7. Product Development and Market Trends for Functional Ingredients

As summarized in Table 4, functional food innovation has rapidly expanded beyond traditional formats to include a wide range of novel strategies aimed at enhancing nutritional value, sensory appeal, and consumer relevance. For example, antioxidant dietary fiber from plant by-products is being used to fortify meat products, addressing both clean-label demands and oxidative stability. Similarly, protein–polyphenol interactions are applied in dairy matrices to improve bioactive delivery and mask undesirable flavors, while mucilage-based fat replacers offer vegan, allergen-free alternatives with added metabolic benefits. Propolis-enriched confectionery and plant-based beverages highlight how functional ingredients can be incorporated into everyday products without compromising taste or convenience. These examples reflect a broader industry trend toward customizing formulations for health impact, consumer expectations, and market competitiveness.

### 7.1. Innovation in Functional Food Products

Functional foods encompass a wide range of categories, including functional beverages, dairy products, meat products, snacks (such as cookies, yogurt, and cereals), fruits, and baked goods [177]. Bioactive peptides derived from various food sources are also gaining significant attention. When incorporated into functional food matrices, they can provide essential amino acids and health benefits. Currently, some bioactive peptides have been commercially produced and are available internationally as ingredients or finished functional food products [39,178].

Various plant materials and their processing by-products can serve as raw materials for antioxidant dietary fiber (ADF), enhancing antioxidant capacity through plant chemicals. They can be used to address dietary fiber deficiencies and mitigate oxidative processes in processed meat products and other foods, while providing health benefits [179,187]. Milk protein, as a multifunctional ingredient in functional foods, offers nutritional value, technical functionality, and health benefits. It can interact with phenolic compounds to enhance delivery or mask astringency/bitterness. Through processing, its functions can be customized, such as protein–phenolic nanoparticles for enhanced delivery or whey protein emulsions for controlled flavor release [180]. Plant mucilage (hydrophilic polymers composed of proteins and water-soluble polysaccharides) extracted from seeds, roots, stems, and fruits exhibits functions such as thickening, gelling, and emulsification, and serves as a fat substitute, providing soluble dietary fiber to support cholesterol/blood sugar management and digestive health. For example, chia seed mucilage (2.5–7.5% *w*/*v*) alters coagulation characteristics in low-fat yogurt while maintaining physicochemical properties, demonstrating its potential as a sustainable, clean-label, and vegan fat substitute [181,188].

### 7.2. Market Trends and Challenges

The global functional food ingredients market is projected to grow from USD 119.2 billion in 2024 to USD 165.8 billion by 2029, at a CAGR of 6.8%. Consumers increasingly prioritize healthy foods and beverages, especially urban populations with rising incomes. An October 2022 Kerry Group survey of 10,000 consumers from 18 countries showed shifting priorities for functional beverages, with 53% interested in scientifically validated ingredients like omega-3, probiotics, and bioactive peptides [189]. Demand for plant-based products with natural bioactives and high bioavailability has surged [185,190]. Advances in materials science, such as engineered colloids (emulsions, foams, and gels), enable innovative designs addressing health and sustainability [124,182]. Berries, widely available and convenient, exemplify plant foods with strong potential as sources of nutraceuticals [60].

Meeting rising global dietary protein demand is challenging, prioritizing sustainable alternatives to animal sources. Insect protein, like EFSA-approved yellow mealworm (*Tenebrio molitor* larvae), is emerging, already common in Africa, Asia, and Australia. The shift to plant-based proteins is driven by sustainability and health benefits, reducing risks of cardiovascular diseases, type 2 diabetes, and cancers. Some functional colloids (nanoemulsions, microcapsules, coated droplets, and nanocellulose) have been used to address obesity-related issues due to their ability to regulate lipid digestion and satiety [184]. For example, commercial products use palm oil or oat oil fractions rich in galactolipids, which release free fatty acids in the distal small intestine, triggering the “ileal brake” mechanism to promote satiety and control food intake [124].

At the same time, consumers in developed markets (such as the United States, Canada, and Europe) are increasingly preferring natural, minimally processed plant-based beverages in pursuit of health and sustainability [186]. Products such as coconut water and maple syrup are rapidly gaining popularity in Western markets due to their natural content of minerals, organic acids, and phytochemicals [183]. Stringent quality and safety standards are required to ensure product consistency and compliance. Combined with issues such as the scarcity of raw materials and the complexity of sourcing natural ingredients, these factors drive up costs. These create two major constraints in the functional food ingredients market: production costs and the substantial investment required [189]. The absence of either of these elements may hinder the development of functional ingredients and limit market growth.

## 8. Clinical Research and Safety Evaluation

### 8.1. The Role of Clinical Trials

Human clinical trials are vital for validating functional food health effects. Studies on diets enriched with marine omega-3s (EPA/DHA) and polyphenols show impacts on lipoprotein metabolism and atherosclerosis markers in high-risk groups, including reduced postprandial lipids in large VLDL, increased IDL cholesterol, triglyceride-enriched LDL, and decreased HDL triglycerides. These interventions also lower oxidative stress (via urinary 8-isoprostanes), improve glucose homeostasis (polyphenol-driven blood glucose reduction and moderated insulin), alter gut hormones (omega-3-reduced GLP-1), and remodel lipids (HDL phospholipid changes as early biomarkers). Recent trials deepen this: a 2024 review highlighted complementary omega-3/polyphenol effects on lipid profiles in CVD patients, potentially enhancing statin therapy.

Natto is often associated with health and longevity among Japanese people [191]. Nattokinase (NK) has been identified as the primary active ingredient in natto and has beneficial cardiovascular effects [192]. Preclinical evidence also suggests potential benefits for Alzheimer’s disease and other conditions. Accumulated evidence suggests NK exerts protective effects primarily through its proteolytic (fibrinolytic) activity. In humans, data support NK’s potential as a thrombolytic/antithrombotic agent. Six randomized trials involving 546 participants have been reviewed, showing that nattokinase supplementation lowered systolic and diastolic blood pressure by roughly 3–4 mmHg without notable adverse events. However, at the relatively low doses studied, it produced no lipid-lowering benefit, in fact, total cholesterol rose slightly and triglycerides were unaffected, underscoring the need for dose–response studies to clarify its efficacy on blood lipids [193].

### 8.2. Safety Evaluation

Before launching functional foods onto the market, strict safety evaluations and supervision are required, especially when it comes to designing new proteins and new delivery methods [160]. Functional food regulations vary widely around the world, with each country having its own relevant food regulations and guidelines. The US FDA does not explicitly classify “functional foods” as a separate category but regulates them as food/supplements under the FFDCA/DSHEA framework, which allows for flexibility in making qualified health claims even with limited evidence, potentially leading to health-related safety issues post-market [194]. In contrast, the EU enforces a stricter regulatory framework under Regulation No. 1924/2006, resulting in high rejection rates and slow innovation under its stringent regulatory standards [195,196]. In Japan, the government has implemented a “two-tier” management system since 1991: FOSHU requires official approval, while FFC only needs to be registered. Traditional ingredients such as nattokinase thus benefit from this [197]. In China, applications must be made to the State Administration for Market Regulation, with 27 specific functions being restricted. Imported products undergo particularly strict reviews. In South Korea and Thailand, the prior approval system is adopted, and it is also aligned with the regulations of the Association of Southeast Asian Nations [198].

Safety assessment is relatively complex. Taking bioactive peptides as an example, peptides derived from safe foods may pose potential risks, and harmful compounds may theoretically be produced during processing and storage, even if no cases have been reported to date. Currently, the most important concern is allergenicity, as a large number of active peptides are derived from highly allergenic raw materials, which have been shown to retain IgE-binding epitopes and cause allergies [199]. Future clinical trials evaluating bioactive peptides should incorporate thorough safety assessments and pharmacokinetic analyses to facilitate translation to real-world applications [200].

Limited human pharmacodynamic and pharmacokinetic data hinder the systematic clinical translation of many bioactive peptides. Robust clinical trials evaluating the metabolic fate, efficacy, and safety of identified bioactive peptides are necessary to substantiate health claims and gain regulatory approval [7]. Safety considerations also apply to novel food technologies like 3D printing, particularly if non-traditional ingredients are used, necessitating strict regulatory oversight. The safety of bioactive compounds incorporated into printed foods must be ensured, considering potential adverse reactions like allergies, toxicity at high doses, or interactions with medications [140]. Even natural products like Manuka honey require safety monitoring for potential contaminants, including heavy metals, pesticide residues, and processing-related compounds like hydroxymethylfurfural (HMF) [86], which need to be controlled within regulatory limits.

## 9. Future Research Directions

Future research in functional ingredients must pursue greater precision, personalization, and sustainability by systematically exploring underutilized sources such as rare plants, marine organisms, microbial metabolites, and insect proteins to discover novel bioactives with targeted health benefits. Integrating multi-omics approaches together with nanotechnology and artificial intelligence will be crucial for understanding complex interactions among ingredients, the microbiome, and the host, for achieving controlled release of bioactives in vivo and for developing predictive models that enable truly personalized nutrition. It will also be important to investigate synergistic combinations of compounds to enhance efficacy and to translate laboratory-scale extraction and formulation methods into scalable, green and cost-effective industrial processes. Success in these efforts will depend on strong collaboration among food scientists, nutritionists, engineers, data scientists, clinicians, and social scientists, as well as on the establishment of unified technical standards, robust validation protocols (particularly for AI-driven discoveries), and clear ethical and regulatory frameworks that foster consumer trust and protect environmental resources.

## 10. Limitations

Although this review comprehensively examines functional ingredients from aspects such as biological activity, technological innovation, and applications in food, the limitations and biases of the conclusions should not be overlooked. In the literature selection, while we prioritized studies from the past five years to ensure timeliness, we may have overlooked the impact of long-term epidemiological research. For instance, in the section on omega-3 fatty acids, although recent meta-analyses on cardiovascular metabolic benefits were cited [45,46,47,48,49], we did not address the increased bleeding risk associated with high-dose supplementation revealed in earlier cohort studies, thereby limiting a comprehensive assessment of the cumulative evidence. Furthermore, our literature selection emphasized beneficial aspects and predominantly relied on mouse and in vitro models, failing to account for potential inconsistencies in efficacy or bioavailability that human trials might reveal [60,61,62,63,64,65,66,67,68,69].

Moreover, the studies were predominantly concentrated in Asian (China, Japan) settings, with a lack of data from regions such as Africa and Latin America. Particularly in the probiotics section, the limited use of examples from typical Asian fermented foods—such as kimchi and mare’s milk wine—overlooked the diversity of gut microbiota resulting from global dietary patterns [54]. Non-systematic retrieval introduced bias in the 3D/4D printing technology field [137,138,139,140,141]. Furthermore, selecting English-language literature as one of the screening criteria may have limited access to a broader range of studies. This underscores the necessity for more balanced and systematic analyses in future research.

## 11. Conclusions

This review highlights the functional components’ role in driving the food-related industries beyond mere nutrition to offer additional health benefits. By examining their molecular structures, physicochemical properties, microbial interactions, and bioavailability, we demonstrated the mechanisms for combating chronic diseases such as short-chain fatty acid production and inflammation regulation. Innovations in areas such as green extraction, microencapsulation, targeted delivery, and 3D/4D printing have enhanced the stability, efficacy, and consumer acceptance of various food matrices. Nevertheless, there are still challenges in terms of the bioavailability, sensory properties, and regulatory compliance of the new proteins and systems. By conducting interdisciplinary research and implementing evidence-based commercialization, these gaps can be bridged, which will drive functional foods to improve global public health, reduce medical costs and promote sustainable production.

## Figures and Tables

**Figure 1 foods-14-03141-f001:**
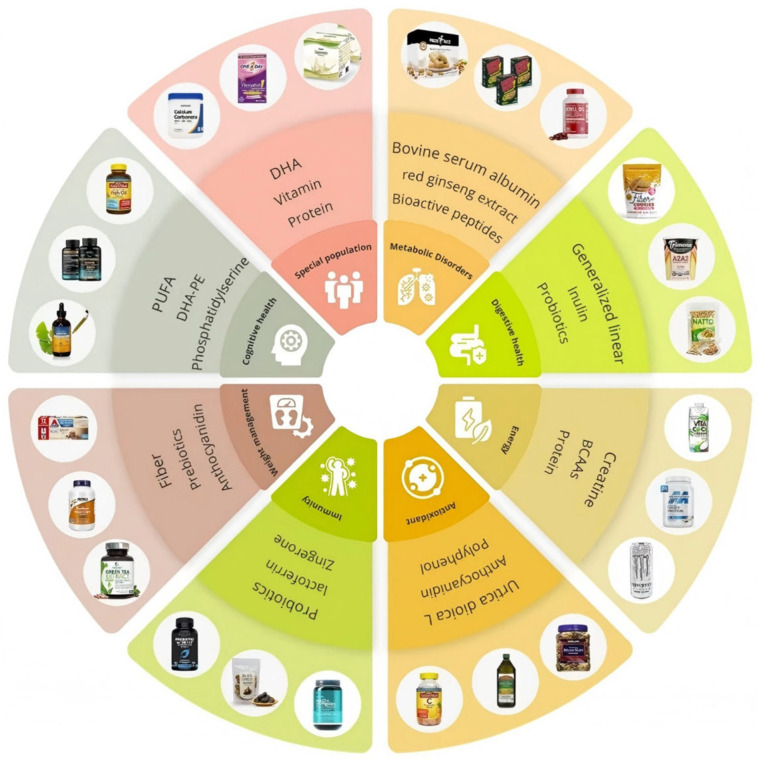
Commercial examples of functional food products categorized by bioactive ingredients and target health benefits.

**Figure 2 foods-14-03141-f002:**
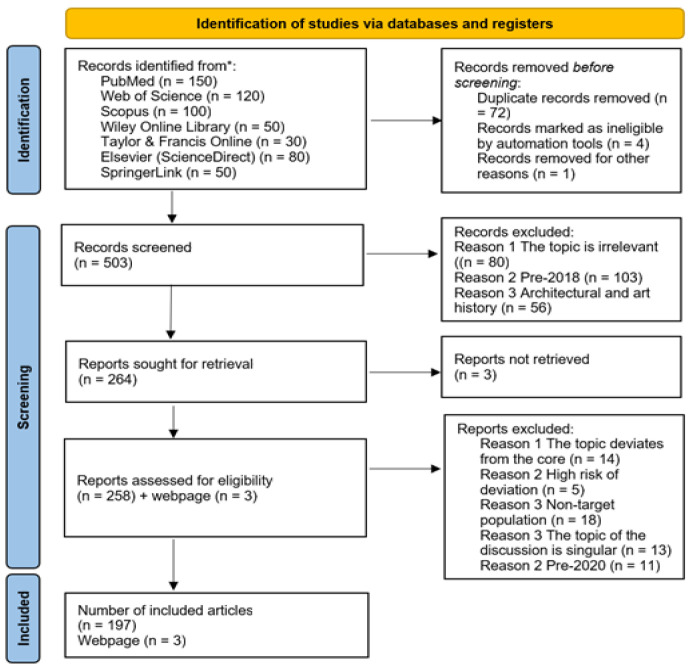
Literature selection diagram. * The numbers in parentheses indicate the number of records initially retrieved from each database before duplicate removal.

**Figure 3 foods-14-03141-f003:**
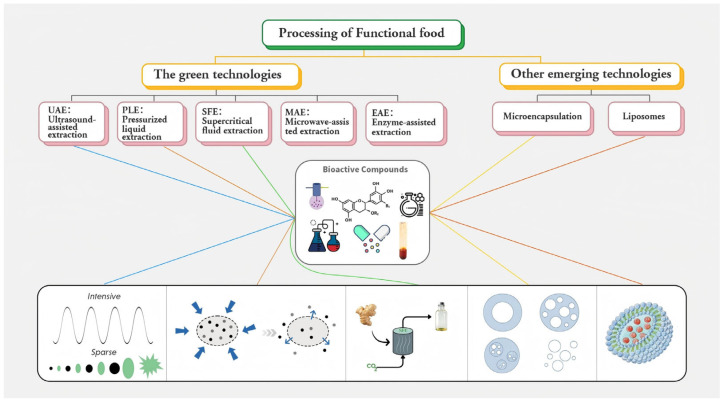
Overview of Green and Emerging Technologies for Functional Food Processing.

**Table 1 foods-14-03141-t001:** Common functional ingredients, their type, source, biological activities, and key study outcomes.

Type	Type	Functional Ingredient	Source	Principal Biological Activities	Key Evidence/Outcomes	Reference
Carbohydrate-based Functional Ingredients	Dietary fiber	*Psyllium husk* (*Plantago ovata*)	Seed coat of *P. ovata*	Modulates bowel function and alleviates IBS pain	A 6-week double-blind RCT in 88 children showed psyllium reduced pain-episode frequency in boys versus placebo, with no effect in girls	[28,29]
		Non-starch polysaccharides (cellulose, hemicellulose, pectin)	Multiple plant tissues	Improves glycemic and lipid profiles, modulates gut microbiota, lowers CVD and T2DM risk	Short-chain fatty acid–mediated gene regulation and anti-inflammatory effects observed	[16,18]
		Curry-leaf polysaccharides (*Murraya koenigii*)	Leaves	Restores gut barrier function, regulates SCFA profile, exhibits anti-obesity effects	Alkaloid fraction from *M. koenigii* alleviated HFD-induced obesity and insulin resistance in mice	[23]
	Prebiotic	Designed dietary-fiber prebiotic	Synthetic soluble fiber blend	Enhances SCFA-producing gut bacteria, improves barrier function, reduces inflammation	10-day clinical study (20 g/day) in Parkinson’s patients decreased zonulin and plasma NfL levels	[32]
		Enzymatically modified pea-peel fiber	Pea by-product	High-solubility DF with prebiotic potential; increased SCFA in vitro	Boost cell biomass and relative growth rate of probiotic bacteria	[31]
Protein-based Functional Ingredients	Proteins and Peptides	Bovine serum albumin (BSA)	Whey	Bovine colostrum whey powder, Amino acids	Whey-derived products offer bioactive peptides, improved functionalities, and diverse food and health applications through optimized processing techniques	[83]
		Food-derived bioactive peptides	Insects, seaweed, and oilseed by-products	ACE inhibition, antihypertensive, antioxidant, immunomodulatory	Emerging protein-derived hydrolysates show promise for managing hypertension, diabetes, obesity, and neurodegeneration	[84]
		Alternative proteins (*Tenebrio molitor*, algal protein)	microalgae	Antioxidant and antimicrobial effects	Offer high-protein, amino acid–rich biomass; functional peptides; and sustainable extraction methods for food and pharma applications	[85]
Fat-based Functional Ingredients	ω-3 PUFA	EPA + DHA (fish oil)	Marine lipids	Lowers triglycerides, reduces inflammation, improves mitochondrial function	Improves lipid profiles, insulin sensitivity, blood pressure, liver function, inflammation, oxidative stress, and mitochondrial performance, and may reduce sudden cardiovascular event risk in diet-related disorders	[45]
		EPA vs. DHA mechanistic differences	Steaming and baking in foil on the retention of EPA and DHA in three fish species	EPA superior in anti-depressant effect; DHA stronger for insulin resistance via GPR120/PPARγ	Steaming fish at 100 °C preserves significantly more EPA and DHA, with retention up to about 60% in Indian mackerel and over 500 mg per 100 g total EPA + DHA in steamed Indian scad, compared to lower retention when baking in foil at 160 °C	[49]
Other Functional Ingredients	Probiotic	*Lactobacillus plantarum* formulation with isatin	*L. plantarum* culture + *Couroupita guianensis* fruit isatin	Antioxidant, antimicrobial, and cholesterol-lowering	In vitro and oral-fluid model showed 67% LDL-C reduction and 98.8% probiotic survivability	[53]
		Common probiotic genera (*Lactobacillus*, *Bifidobacterium*, *S. boulardii*)	Fermented foods (yogurt, kimchi, kefir)	Strengthens mucosal barrier, increases IgA, and reduces intestinal inflammation	Gut health by enhancing barrier function, immunity and reducing inflammation; *Bifidobacterium* raises IgA levels and cuts harmful bacteria in the gut	[54]
	Phenolic compound	6-Gingerol	*Ginger rhizome*	Provides hepatoprotective and anti-inflammatory effects	In DEN-induced liver-injury rats, 50 mg/kg reduced ALT, AST, and ALP, and mitigated tissue damage	[65]
		Zingerone	*Ginger rhizome*	Antioxidant, anti-inflammatory, anti-apoptotic	Zingerone pretreatment in lead-exposed rats improved antioxidant enzymes and protected organs	[66]
		Quercetin, catechins	Tea, fruit, veg.	Geroprotective; interact with ageing pathways	Polyphenol-rich diets of Blue Zone centenarians may promote healthy ageing and longevity by targeting conserved biological mechanisms to reduce age-related disease risk	[64]
		Manuka honey polyphenols	Manuka honey	Antimicrobial, antioxidative, wound healing	Demonstrated antioxidant, anti-inflammatory, immunomodulatory, antimicrobial, and anticancer effects	[86]
Combination therapy		Red-ginseng extract (RGEP) + red-ginseng dietary fiber (RGDF)	*Panax ginseng* root and fiber	Gut-barrier protection; anti-inflammatory	RGEP and RGDF supplementation in DIO mice (4–8 weeks) reduced markers of inflammation and intestinal permeability, including α-1-antitrypsin, CRP, iNOS, NF-κB, MPO, calprotectin, urinary indican, and β-glucuronidase	[72]
		Cranberry polyphenols + EPA/DHA	Cranberry juice (200 mL) + fish-oil capsule (180 mg EPA + 120 mg DHA, BID)	Glycemic control; cardiometabolic and periodontal benefit	An 8-week study in diabetic individuals with periodontitis showed decreased HbA1c, increased HDL-C, and better periodontal indices	[77]

IBS; irritable bowel syndrome; RCT, randomized controlled trial; CVD, cardiovascular disease; T2DM, type 2 diabetes; SCFA, short-chain fatty acid; HFD, high-fat diet; NfL, markers of neurodegeneration; DF, dietary fiber; ACE, angiotensin-converting enzyme; EPA, eicosatetraenoic acid; DHA, docosahexaenoic acid; LDL-C, low-density lipoprotein cholesterol; IgA, immunoglobulin A; DEN, diethylnitrosamine; ALT, alanine aminotransferase; AST, aspartate aminotransferase; ALP, alkaline phosphatase; DIO, diet-induced obesity; CRP, c-reactive protein; iNOS, inducible nitric oxide synthase; MPO, myeloperoxidase; BID, The Latin abbreviation “bis in die” means “twice a day”; HDL-C, High-density lipoprotein cholesterol; HbA1c, hemoglobin A1c.

**Table 2 foods-14-03141-t002:** Technologies and strategies that boost the bioavailability, stability, and personalization of functional ingredients.

Theme/Tool	Main Purpose	Core Mechanism and Key Insight	Representative Bioactives Involved	Typical Food or R&D Application	Key Reference
Bioavailability fundamentals	Disease prevention and therapeutic support	Modulate inflammation, oxidative stress, lipid metabolism, gene expression; exert antimicrobial and anticancer effects	Polyphenols, carotenoids, glucosinolates, terpenes, alkaloids, ω-3 PUFA, CLA, chitosan, probiotics, marine bioactives	Functional foods, dietary supplements, drug development	[87]
Fermentation of dietary fiber	Enhances the release of bound phenolics and increases their solubility	Microbial enzymes cleave cell wall matrices, creating shorter, more fermentable fragments	Sweet-potato, cereal, or legume DF	Fermented DF powders for gut-health beverages	[90]
Micronization of DF	The increased surface area promotes greater release and absorption of polyphenols and carboxyl groups	Jet-milling reduces particle size (<50 µm), exposes functional moieties	Fruit and veg by-product fibers	High-fiber smoothies, bakery powders	[13]
Excipient foods	Act as co-ingested carriers that enhance uptake of lipophilic or labile actives	Add digestible lipids, binding proteins, or permeability enhancers to promote micellization and protect against oxidation	Curcumin, quercetin, carotenoids, CoQ10	“Bioavailability-boost” snack bars, shots	[96]
Green extraction (UAE, MAE, EAE, PLE, SFE)	Obtains high-purity actives while lowering solvent, time and thermal damage	Ultrasound/microwave cavitation, enzyme hydrolysis, pressurized liquids, supercritical CO_2_	Polyphenols, carotenoids, essential oils, peptides	Clean-label extracts for functional beverages, gummies, capsules	[123]
Supercritical CO_2_ (SFE)	Green extraction of raspberry seed oil e	SFE achieved comparable yield to Soxhlet, but higher ω–3 FA content; optimized via Box–Behnken design; better oil quality and full material exhaustion	ω–3 fatty acids, unsaturated fatty acids, tocopherols	Clean-label lipid ingredients for functional foods, nutraceutical oils	[132]
Microencapsulation	Enhances probiotic stability and viability during processing and digestion	Provides a protective matrix that shields probiotics from heat, oxygen, and acidity; enables targeted release in the GI tract	*Lactobacillus*, *Bifidobacterium strains*	Functional dairy foods (e.g., pasta filata cheese), dietary supplements	[104]
Enzymatic hydrolysis	Production of functional food ingredients	Catalyzing the hydrolysis of peptide bonds and releasing functional food components such as bioactive peptides	Peptide	Dairy, meat, plant, seafood	[113]
Nano-vesicular carriers (NVCs)	Improve stability, bioavailability, and functionality of bioactives in foods	NVCs (e.g., liposomes, niosomes, phytosomes, and transfersomes) enhances antioxidant capacity, control release at various pH/storage conditions, reduce cytotoxicity, improve digestibility, mask taste/smell, inhibit biofilm gene expression, and maintains sensory properties in fortified foods	Anthocyanins, d-limonene, tannic acid, ω-3 fatty acids, iron	Omega-3 fortified juices, powdered drink mixes; Functional foods with antioxidant, anti-inflammatory, or sensory-sensitive bioactives	[102]
3D/4D food printing	Personalized nutrition; time-dependent shape or flavor change	Extrusion of stimuli-responsive biopolymers; programmable geometry	Fenugreek-gum + flax protein toddler snacks; protein-rich insect pastes	Tailor-made textures for dysphagia, space-saving packaging	[141]
Molecular simulation (MS)	In silico prediction of binding, stability, ADME–Tox	Atomistic docking and MD reveal enzyme inhibition, receptor targeting	*C. papaya* polyphenols vs. α-amylase/α-glucosidase	Accelerates lead-ingredient screening before wet lab	[144]
Artificial intelligence (AI/ML)	Accelerate ingredient development for improved sensory quality	Use of AI models to classify compounds by taste category and uncover structural determinants of taste; supports predictive organoleptic profiling	Mediterranean bioactives, phytochemicals, synthetic flavorants	AI-enabled R&D for functional food, nutraceuticals, and reformulated products	[143]
AI-driven fraud detection	Identify food adulteration and preserve consumer trust	Combines chemical fingerprints with machine learning to detect anomalies or counterfeit patterns; addresses issues driven by financial motives and geopolitical instability in supply chains	Authentication markers (e.g., isotopic, molecular)	Anti-fraud testing for raw/processed coffee, cocoa, and tea	[148]
AI and digital tools	Enable intelligent ingredient discovery and process control	AI predicts protein–function relationships, identifies health-promoting molecules, supports real-time quality monitoring through sensor-integrated digital platforms	Predictive models for peptides, flavor precursors	Smart formulation tools, health-enhancing plant-based foods	[150]

PUFA, polyunsaturated fatty acid; CLA, conjugated linoleic acid; DF, dietary fiber; CoQ10, coenzyme Q10; UAE, ultrasound-assisted extraction; MAE, microwave-assisted extraction; EAE, enzyme-assisted extraction; PLE, pressurized liquid extraction; SFE, supercritical fluid extraction; FA, fatty acid; GI, gastrointestinal; MD, molecular dynamics; ADME, absorption, distribution, metabolism, excretion.

**Table 3 foods-14-03141-t003:** Sensory attributes of functional-ingredient products and determinants of consumer acceptance.

Focus Area	Core Insight	Practical Implication for Product Developers	Reference
Advanced sensory evaluation tool-box	Traditional discrimination, descriptive, and hedonic tests are now complemented by biometrics (facial EMG, HR, GSR, eye-tracking), virtual/augmented reality (VR/AR) and electronic-nose/e-tongue systems.	Combines objective physiological data with self-report, allows testing in immersive, context-rich environments and shortens time-to-insight.	[153]
VR-enabled panels	VR can synchronously manipulate sight, sound, odor, haptics, and taste. Immersive scenes reshape liking and purchase intent.	Prototype “farm-to-table” or “carbon-neutral” contexts to quantify eco-premium consumers are willing to pay.	[158]
Bitterness of bioactive peptides	5–6 human T2R receptors respond strongly to food-derived peptides; bitterness limits acceptance.	Spray-dry encapsulation with maltodextrin/cyclodextrin, exopeptidase trimming, or QSAR-guided sequence selection masks bitterness while preserving bioactivity.	[7]
Flavor pairing science	Most foods/beverages are co-consumed; sensory synergy or suppression drives overall liking.	Optimize functional beverages with compatible carriers (e.g., plant sterol juice + citrus notes) to avoid off-flavors.	[159]
Sociodemographic drivers	Age, gender, education, household type, nationality, and marital status modulate acceptance. Nordic and Finnish consumers show higher readiness than US/Danish counterparts; Chinese > German for health-claim products.	Segmentation is essential—tailor message/format to demographic clusters.	[160]
Message framing in DTC genetic testing	Message features (e.g., sidedness, hedging) and prior experience with genetic tests significantly affect trust, information processing, risk perception, and attitudes.	To boost purchase intention, companies should enhance trust in both message and brand by using two-way refutational messaging and hedging, especially for experienced users.	[161]
Health motivation and self-efficacy	Preventive-health orientation, self-efficacy and self-esteem are top internal motivators.	Emphasize tangible health outcomes (e.g., cholesterol reduced by X%) and empower consumers with usage guidance.	[162]
Adoption of autonomous shuttles	Perceived usefulness and enjoyment drive perceived value, which predicts adoption intention; perceived risk has no significant effect.	Developers should prioritize enhancing perceived usefulness and enjoyment of autonomous shuttles to increase public adoption, especially in emerging markets.	[163]
Green advertising credibility	Eco-brand familiarity boosts ad credibility when the product is inherently green; green purchasing orientation can lower ad credibility, especially for low-cost items.	To strengthen ad effectiveness, brands should build familiarity and launch truly green products; avoid overemphasis on claims for low-cost items that may trigger skepticism.	[164]
Social proof and herd behavior in food choice	Social proof (e.g., reviews/ratings) significantly influences food choices, while herd behavior and influential sources have limited effect; health preference can override social influence.	Use consumer reviews/ratings as persuasive tools to promote healthy products; focus less on influencer marketing and more on visible peer-generated feedback.	[165]

EMG, electromyography; HR, heart rate; GSR, galvanic skin response; T2R receptors, type 2 taste receptors; QSAR, quantitative structure–activity relationship; DTC, direct-to-consumer.

**Table 4 foods-14-03141-t004:** Emerging strategies in functional food innovation: technical insights and market relevance.

Theme	Core Focus	Key Technical/Nutritional Insight	Market Relevance	Reference
Functional food categories	Functional meat, beverages, dairy, fruit products and snacks (biscuits, yoghurts, cereals)	Demonstrates the breadth of matrices that can deliver bioactives without compromising sensory quality	Signals continued diversification of “every-day” products carrying health claims	[177]
Bioactive peptides	Peptides from diverse raw materials	Several peptides already commercialized; incorporated into bars, drinks, dairy	Peptide-fortified foods add scientifically validated benefits beyond basic nutrition	[178]
Antioxidant dietary fiber (ADF) in meat	Plant by-products rich in DF + polyphenols	ADF improves fiber content and retards oxidative spoilage in meat	Meets demand for “clean-label” meat products with added health halo	[179]
Milk-protein functionalization	Whey/casein combined with polyphenols or volatiles	Protein–polyphenol nanoparticles and whey-based emulsions enhance delivery, mask bitterness	Customizable protein systems widen application in high-protein functional foods	[180]
Mucilage as vegan fat replacer	Plant-derived hydrophilic polymer (chia, basil, okra, etc.)	7.5% chia mucilage yogurt reduced syneresis, preserved texture, and offered dietary fiber-related benefits on cholesterol and glycemic control	Clean-label, sustainable, allergen-free alternative to dairy or animal fat	[181]
Propolis in confectionery	Stingless-bee propolis chewing gum	Clinical studies in children demonstrated a reduction in Streptococcus mutans biofilm formation, an increase in salivary calcium and phosphate concentrations, and a decrease in dental plaque accumulation	Demonstrates delivery platform for antimicrobial phytochemicals; opens oral-health snacking niche	[182]
Plant-based functional beverages	Coconut water, maple sap, minimally processed drinks	Consumers prefer high bioavailability of native phytochemicals	Growth of “natural and barely processed” beverage segment in US/CA/EU	[183]
Engineered food colloids	Nano-/micro-structured emulsions, nanocellulose	Enables healthier, tastier, safer, more sustainable foods	Colloidal design tackles reformulation challenges (fat, salt, sugar reduction)	[184]
Berries as nutraceuticals	Polyphenol-rich berries	High convenience and palatability	Rising demand for ready-to-eat antioxidant sources	[185]
Clean-label plant beverages	Coconut water, maple sap, low-processed juices	“Raw” nutrient profile, natural antioxidants retained	Younger consumers value authenticity, low processing, and combined energy-drink formats	[186]

DF, dietary fiber; US, the United States; CA, Canada; EU, European Union.

## Data Availability

No new data were created or analyzed in this study. Data sharing is not applicable to this article.

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
