# Peer review of "Functional Ingredients: From Molecule to Market—AI-Enabled Design, Bioavailability, Consumer Impact, and Clinical Evidence"

_foods, 2025, doi:10.3390/foods14173141_

Round 1

Reviewer 1 Report

Comments and Suggestions for Authors

In today's world where nutritional patterns are changing rapidly and functional food ingredients have become a trend as well as meeting specific health needs, this study can be considered as a study that will contribute to the literature with its subject and content. In this context, I recommend that the study be re-evaluated by making significant revisions, especially regarding its content and categorization.

I have some comments/suggestions:

# The article is very understandable and well written. The authors have effectively utilized current and relevant sources, demonstrating a strong understanding of the subject matter.

Title: The title of the article is quite clear and sufficient in explaining the purpose and content of the study and in presenting an updated evaluation along with previous studies.

Keywords: The important consideration when selecting keywords is to include terms that highlight the study’s additional significant findings or key concepts, rather than simply repeating words already used in the title. This approach enhances the discoverability of the article and allows it to be indexed and reviewed from a broader perspective.

Authors should be mindful of this and revise some recurring keywords.

Introduction: The introductory section of the study is generally well designed. The purpose of the study is stated without being too long or including unnecessary details, and the scope of the topics to be discussed is limited.

# The first thing I'll suggest in this article is to reorganize some of the main headings. Examining functional ingredients under three main headings—carbohydrate, protein, and fat-based—will allow you to make a much more fundamental classification. Because the functional food ingredients you present here as dietary fibers also fall under the subheadings of prebiotics.

# As I mentioned, reorganizing Table 1 according to this classification will present the studies with a much more accurate, clear and understandable classification.

# Table 2, which also lists technologies and strategies for enhancing the bioavailability, stability, and customization of functional ingredients, makes no mention of enzymatic hydrolysis, which is used in protein modification. As is well known, enzymatic hydrolysis is one of the most important processes used in the production of specialized and specialized functional food ingredients, such as bioactive peptides. It should definitely be included as one of the subheadings under the heading "Innovative Technologies and Applications." It is mentioned very briefly in biactive peptides, but it should definitely be included under a subheading as I suggested.

# (5.2) It would be much more explanatory to detail the acceptance of functional food ingredients based on regional and cultural differences with some important examples. For example, although the consumption of certain insects as protein sources is quite common in Far Eastern cultures, and this constitutes a significant market for functional ingredients, it can be mentioned that these do not receive sufficient attention in some cultures due to cultural and religious reasons.

# Some sections of the article have included too many subheadings. For example, 2.1.6. Synergistic Effects: Bioactive Peptides. This heading doesn't need much context. These proteins and bioactive peptides can be mentioned in the final paragraph.

# Under the section titled Market Trends and Challenges, it would enhance the clarity and depth of the discussion to include specific data on the market share and volume of functional food ingredients within the global food industry. Additionally, identifying and briefly discussing the key functional ingredients driving this market—such as probiotics, prebiotics, peptides, or plant-based bioactives —would help present the topic in a more structured and comprehensive manner, thereby strengthening the overall narrative.

# The meaning of abbreviations should be stated at the first place they are mentioned in the text (Line 38, 120)

# Even if the long form of the abbreviations given in the tables is given in the text, the long form must be explained again as a footnote under the tables. Please check all tables and add relevant abbreviations.

Good luck with corrections…

Reviewer 2 Report

Comments and Suggestions for Authors

Comments on the manuscript:
1. The title could be more appealing to increase potential reader interest.
2. A clear methodology section should be included that describes the approach followed in writing the review article (this could be section 2). In addition, MDPI uses the PRISMA methodology in this type of manuscript, so it should be formally incorporated, accompanied by a figure representing the corresponding flowchart to illustrate the review article's structure.
3. It is necessary to detail the total number of articles consulted in the new section 2, the inclusion and exclusion criteria applied, as well as the databases used (e.g., Scopus, Wiley, Taylor and Francis, Elsevier Springer, Web of Science, PubMed, etc.). In addition, it is suggested that the period of analysis be limited preferably to the last five years, to ensure that the information is up to date (2020-2025).
4. It is recommended that an Excel spreadsheet containing the complete list of articles reviewed, as well as the final sample selected (the 211 articles used for the critical analysis and/or meta-analysis), be attached as a supplementary file.
5. The first paragraph of the introduction should be reworded. It currently lacks argumentative force. We suggest starting with a specific problem that captures the reader's attention and directly introduces the importance of the topic being reviewed. The introduction should also clearly highlight what this review contributes compared to previous reviews. It is important to establish the substantial difference or comparative advantage that justifies its necessity and originality.
6. Illustrative figures should be included to complement the concepts developed throughout the text. We suggest using tools such as BioRender or similar ones to improve visual clarity and the reader's experience.
7. Table 1 contains valuable information but is poorly integrated into the text; we recommend referencing and commenting on it within the body of the article (consider doing the same with the other tables).
8. The manuscript describes clinical studies superficially and requires a critical comparative analysis of results and limitations.
9. The regulatory section is weak; an analysis of international differences (FDA, EFSA, Asia) is lacking.
10. The section on AI and molecular modeling is too general and should delve deeper into practical limitations (datasets, biases, experimental validation).
11. The discussion of emerging technologies does not adequately address scalability, costs, or energy efficiency.
12. References to negative interactions or antagonisms between bioactives, not just synergies, are missing.
13. Some sections repeat ideas (e.g., bioaccessibility and bioavailability), which creates redundancy.
14. Several statements on emerging technologies lack recent citations and should be updated.
15. The discussion of techniques should include comparative data with other methods.
16. It is valuable to incorporate a specific section that analyzes the industrial viability of the technologies or strategies mentioned. This could address aspects such as costs, scalability, impact on product quality, and real applicability.
17. Some very dense paragraphs need to be polished to make them interesting to an audience that is not necessarily familiar with the subject.
18. It is recommended to include a specific section on conclusions; it is suggested to summarize the key ideas in compact paragraphs. The limitations of the study and possible biases in the selection of literature have not been included, which is important in a review.
19. It is suggested that a list of abbreviations be added at the end of the manuscript, just before the references section, in the format commonly used in the FOODs journal.
20. The language is redundant in several paragraphs; a style review is recommended to avoid repetition and improve fluency. Forty-five pages seem very long.
21. It is recommended to reduce the iThenticate similarity index (16%), especially in the materials and methods and results and discussion sections.

Round 2

Reviewer 1 Report

Comments and Suggestions for Authors

The article has become a much more organized and fluent text with its new form. The authors have considered the suggestions very carefully and meticulously. They have significantly improved the work compared to the previous version through an extensive revision process. The aim and scope of the study are well detailed and the boundaries are clearly stated, and the need for different approaches for future studies is expressed. All things considered, I recommend that the article be accepted for publication after correcting a few minor oversights.

I have some comments/suggestions detailed as listed below:

# The Methodology section added to the study strengthens the study as a compilation study, demonstrating its scope, transparency, and ensuring its reproducibility. All things considered, I recommend that the article be accepted for publication after correcting a few minor oversights.

# In the section titled Market Trends and Challenges, the addition of specific data regarding the market share and volume of functional food ingredients in the global food industry has supported the article with more concrete data.

# The relevant sections under the headings discussed in the article are quite well-discussed. I think a few more details could be included here, particularly regarding perbiotics. This section alone can be considered a bit superficial; it could be elaborated on in just a sentence or two.

# The title is acceptable and sufficient to express the scope and objectives of the study.

# Authors must italicize species names. Please, check and correct throughout the text (Line 241), Table 1 (Murraya koenigii , Plantago ovata etc.)

# Please use a standard font for all titles. Additionally, follow title case formatting—only capitalize the first letter of each major word (e.g 3.5. The interaction of functional components Line 282)

# Please check and use a standard font for each section (Table 2 (Enzymatic Hydrolysis)

#Adding the long forms of abbreviations in the tables makes the table contents easier to read. However, I don't think adding the ‘‘Table 1 Abbreviation Notes:’’ statement is necessary. Simply adding the long forms of the abbreviations below the table would be sufficient.

Good luck with corrections…

Reviewer 2 Report

Comments and Suggestions for Authors
  • The authors indicate that they have attached an Excel file with the complete list of reviewed articles, but this file is not included in the manuscript. In addition, there is a discrepancy between the 211 articles initially mentioned and the 200 finally reported, which requires justification.
  • The inclusion of illustrative figures was requested, but the authors decided not to add them. The manuscript lacks visual elements that facilitate the understanding of complex concepts, which weakens its clarity.
  • The authors claimed to have added a list of abbreviations, but no such section appears in the revised document. This contradicts what was stated in the response.
  • Although stylistic adjustments were made, the manuscript remains lengthy (≈47 pages) and redundant in several sections. The authors' justification does not eliminate the problem pointed out by the reviewer. The text remains very technical and challenging for non-specialist readers. It still requires stylistic polishing.
  • The authors included comments on industrial feasibility, but in a scattered manner. No specific section or comparative table was created, which detracts from the clarity of this fundamental aspect.
  • The limitations and biases of the review are mentioned superficially and without concrete examples, which detracts from the strength of this part of the manuscript.
